# Debiasing Made State-of-the-art: Revisiting the Simple Seed-based Weak Supervision for Text Classification

**Chengyu Dong**      **Zihan Wang**      **Jingbo Shang**[*]

University of California San Diego

{cdong, ziw224, jshang}@ucsd.edu

## Abstract

Recent advances in weakly supervised text classification mostly focus on designing sophisticated methods to turn high-level human heuristics into quality pseudo-labels. In this paper, we revisit the seed matching-based method, which is arguably the simplest way to generate pseudo-labels, and show that its power was greatly underestimated. We show that the limited performance of seed matching is largely due to the label bias injected by the simple seed-match rule, which prevents the classifier from learning reliable confidence for selecting high-quality pseudo-labels. Interestingly, simply deleting the seed words present in the matched input texts can mitigate the label bias and help learn better confidence. Subsequently, the performance achieved by seed matching can be improved significantly, making it on par with or even better than the state-of-the-art. Furthermore, to handle the case when the seed words are not made known, we propose to simply delete the word tokens in the input text randomly with a high deletion ratio. Remarkably, seed matching equipped with this random deletion method can often achieve even better performance than that with seed deletion. We refer to our method as *SimSeed*, which is publicly available [1].

## 1 Introduction

Recently, weakly supervised text classification, because of its light requirement of human effort, has been extensively studied (Mekala and Shang, 2020; Wang et al., 2021; Zhao et al., 2022; Meng et al., 2020; Zhang et al., 2021; Meng et al., 2018; Tao et al., 2015; Park and Lee, 2022). Specifically, it requires only high-level human guidance to label the text, such as a few rules provided by human experts that match the text with the labels. These labels, which are not necessarily correct and are

thus often dubbed as pseudo-labels, are then employed to train the text classifier following a standard fully supervised or semi-supervised training framework. State-of-the-art methods mostly focus on designing sophisticated human guidance to obtain high-quality labels, through contextualized weak supervision (Mekala and Shang, 2020), prompting language models (Meng et al., 2020; Zhao et al., 2022), clustering for soft matching (Wang et al., 2021), and complicated interactions between seeds (Zhang et al., 2021).

In this paper, we revisit the seed matching-based weak supervision (denoted as **Vanilla**) (Mekala and Shang, 2020; Meng et al., 2018; Tao et al., 2015), which is arguably the simplest way to generate pseudo-labels, and show that its power was greatly underestimated. Specifically, this simple method matches input text with a label if the user-provided seed words of this label are contained in the input text. For example, in sentiment analysis, a document will be labeled as "positive" if it contains the word "happy". A text classifier is then trained based on all these pseudo-labels.

One can expect a non-trivial number of errors in the seed matching-based pseudo-labels. In an ideal case, if we can select only those correct pseudo-labels for training, the accuracy of the text classifier can be significantly boosted. For example, on the 20 Newsgroups dataset, ideally with only those correct pseudo-labels one can get an accuracy of $90.6\%$, compared to $80.1\%$ obtained on all pseudo-labels (see more in Section 4). In practice, to select those correct labels, a common way is to use the confidence score of a classifier trained on pseudo-labels (Rizve et al., 2021). However, those high-confidence pseudo-labels may not be correct in the weakly-supervised setting, likely because the classifier may fail to learn reliable confidence on these noisy pseudo-labels (Mekala et al., 2022).

In this paper, we take a deep dive into this problem and find that, surprisingly, the high noise rate

---

[*] Corresponding author.

[1] https://github.com/shwinshaker/SimSeed

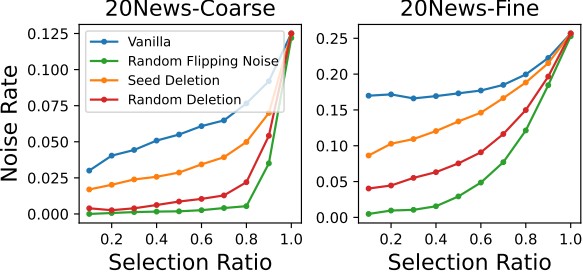

Figure 1: Noise rate in the subset of pseudo-labels selected based on the confidence score of a classifier trained on the pseudo-labeled data. We show the noise rates at multiple selection ratios. ("Vanilla"): the classifier is trained on the original pseudo-labeled data given by seed matching. ("Random Flipping Noise"): the classifier is trained on noisy pseudo-labeled data synthesized using the ground-truth labels, where the data examples, the overall noise rate, noise rate in each class, and the noise transition rate between any two classes are identical to the label noise induced by seed matching. ("Seed Deletion"): The classifier is trained on pseudo-labeled data where the seeds are deleted from the text. (Random Deletion): The classifier is trained on pseudo-labeled data where the words in the text are randomly deleted.

among the pseudo-labels is often not an obstacle to learning reliable confidence at all. In fact, on a set of synthesized pseudo-labels where the noise rate is exactly the same as those given by seed matching, but the noisy labels are generated by **randomly flipping** true labels into other classes, the confidence learned by a classifier can genuinely reflect the correct labels, as shown in Figure 1.

Therefore, we argue that the poor confidence learned on realistic pseudo-labels is largely attributed to the strong but likely erroneous correlation between the text and pseudo-label injected by the seed-matching rule, which we refer to as *label bias*. Such a bias can be easily learned by the text classifier upon training, thus yielding spuriously high confidence on any text matching the seed word and ruining the pseudo-label selection.

To defend against such a label bias, we propose to simply delete the seed words present in the text upon training a classifier on the pseudo-labeled data, which effectively prevents the classifier from learning the biased correlation between seeds and the corresponding pseudo-labels. As shown in Figure 1, such a simple **seed deletion** method can significantly improve the confidence score of the trained classifier and thus help select pseudo-labels with fewer label errors at every selection ratio. Empirical results verify that these

less noisy pseudo-labels can indeed improve the classification accuracy significantly, making seed matching-based weak supervision on par with or sometimes even better than the state-of-the-art.

We further investigate the scenario where the seed words are *not* made known. We propose to delete every word token in the input text randomly and independently. This simple **random deletion** method can improve confidence learning even more as shown in Figure 1. Our theoretical analysis also shows that this random deletion method can mitigate the label bias with a high probability and therefore recover the seed deletion in effect. It is worth noting that both of these methods introduce no additional hyperparameters.

In summary, our contributions are as follows.

- We revisit the seed matching-based weak supervision and find that its effectiveness is mainly limited by the label bias injected by the seed-matching rule.
- We show that simply deleting seed words from the pseudo-labeled texts can significantly alleviate the label bias and improve the confidence estimation for pseudo-label selection, as well as end-to-end classification accuracy achieved by seed matching, on par with or even better than the state-of-the-art.
- We further propose the random deletion method to handle the case when the seed words are unknown and demonstrate its effectiveness both empirically and theoretically.

## 2 Preliminaries and Related Work

**Seed matching as simple weak supervision.** Seed matching (Meng et al., 2018) is probably one of the simplest weak supervision. Specifically, for each label class, a user provides a set of seed words that are indicative of it. A given document is annotated as the label class whose seed words appear in the document, or annotated as the label class whose seed words appear most frequently if multiple such label classes exist. Sophisticated weak supervisions have also been proposed to generate pseudo-labels with better quality, such as meta data (Mekala et al., 2020), context (Mekala and Shang, 2020), sentence representation (Wang et al., 2021), predictions of masked language models (Meng et al., 2020) and keyword-graph predictions (Zhang et al., 2021).

**Confidence learning matters for pseudo-label selection.** Since label errors prevail in the pseudo-labels generated by weak supervision, it is often

necessary to select the pseudo-labels before incorporating them into training (Wang et al., 2021; Mekala et al., 2022). A common method to select those pseudo-labels is to use the model confidence, namely the probability score associated with a deep classifier's prediction, to determine whether a given pseudo-label is correct or not (Guo et al., 2017). However, such confidence often cannot genuinely reflect the correctness of the pseudo-label generated by weak supervision, in that pseudo-labels with high confidence are not necessarily correct (Mekala et al., 2022)

**Backdoor attack and defense.** A problem related to seed matching is the backdoor attack for text classification based on trigger words (Dai et al., 2019; Kurita et al., 2020; Chen et al., 2021). Such attacks corrupt a dataset by inserting a particular word (*i.e.*, trigger) into several documents and change their corresponding labels as ones specified by the attacker. A model fine-tuned on such a dataset will predict the label specified by the attacker whenever a document contains the trigger word. This is largely because the model would overfit the malicious correlation between the trigger word and the specified label, which is similar to the problem when learning pseudo-labels generated by seed matching. Therefore, trigger-based backdoor attacks may be defended by the methods proposed in this work as well, especially random deletion since the attacker will not reveal the trigger word.

## 3 Method

### 3.1 Seed deletion

We describe the details of seed deletion. Specifically, we denote an input document composed of a set of words as $x = \{t_1, t_2, \cdots, t_n\}$ and its pseudo-label given by seed matching as $\tilde{y}$. We denote the set of seed words from class $y$ as $\mathcal{S}_y$. Now for each document in the pseudo-labeled dataset, we generate a corrupted document $\hat{x}$ by deleting any seed word in $x$ that is associated with its pseudo-label, namely $\hat{x} = \{t | t \in x, t \notin \mathcal{S}_{\tilde{y}}\}$. We then train a classifier $\hat{\theta}$ on the corrupted dataset $\hat{D} = \{(\hat{x}, \tilde{y})\}$ and use its confidence score at the pseudo-label $P_{\hat{\theta}}(\tilde{y}|x)$ as an uncertainty measure to select the correct pseudo-labels.

Note that when generating the uncertainty measure on a (document, pseudo-label) pair, one can either evaluate the classifier on the original document or the corrupted document. Empirically we found that evaluating the classifier on the original

document would produce a minor gain.

### 3.2 Random deletion

In real-world applications, the seed words provided by the user may not always be accessible due to privacy concerns or simply because they are lost when processing and integrating the data. We show that it is still feasible to perform seed deletion in a probabilistic manner without knowing seed words, while remaining effective for confidence-based pseudo-label selection.

To achieve this, in fact, we only have to delete the words randomly and independently in a given document. Specifically, give a deletion ratio $p$, for every document $x = \{t_1, t_2, \cdots, t_n\}$, we randomly sampled a few positions $\mathcal{M} = \{i_1, i_2, \cdots, i_{\lceil pn \rceil}\}$, where $\lceil \cdot \rceil$ denotes the ceiling of a number. We then generate a corrupted document by deleting words at those positions, namely $\hat{x} = \{t_i | i \in \{1, 2, \cdots, n\}, i \notin \mathcal{M}\}$. Now on the corrupted dataset $\mathcal{D} = \{(\hat{x}, \tilde{y})\}$, we can train a classifier $\hat{\theta}$ and utilize its confidence score $P_{\hat{\theta}}(\tilde{y}|x)$ for pseudo-label selection, similar to seed deletion.

**Random deletion as a probabilistic seed deletion.** Despite its simplicity, we show with high probability, random deletion can mitigate the label bias induced by seed matching. The intuition here is that since the document only contains one or a few seed words, by random deletion it is very likely we can delete all seed words while retaining at least some other words that can still help learning.

Specifically, we consider a particular type of corrupted document $\hat{x}$ that contains no seed word, *i.e.*, $\hat{x} \cap \mathcal{S}_{\tilde{y}} = \varnothing$, but contains at least one word that is indicative of the true class, *i.e.*, $\hat{x} \cap \mathcal{C}_y \neq \varnothing$, where $\mathcal{C}_y$ denotes the set of words that are indicative of the class $y$. Since such a document no longer contains the seed word, its pseudo-label is not spuriously correlated with the text. At the same time, it contains class-indicative words that can help the classifier learn meaningful features.

We then investigate the probability that a document becomes such a particular type after random deletion. We term such probability as the seed-deletion rate $r_{\text{SD}}$, which is defined as

$$r_{\text{SD}} := P(\mathbf{1}(\hat{x} \cap \mathcal{S}_{\tilde{y}} = \varnothing, \hat{x} \cap \mathcal{C}_y \neq \varnothing)), \quad (1)$$

where $\mathbf{1}(\cdot)$ is the indicator function. In an ideal case where $r_{\text{SD}} = 1$, we can completely recover the effect of seed deletion on eliminating label bias.

Now since each word in the document is independently deleted, we have

$$r_{\text{SD}} = p^{n_s} \cdot (1 - p^{n_c}), \qquad (2)$$

where $n_s := |\mathcal{S}_{\tilde{y}}|$ denotes the number of seed words in the document and $n_c := |\mathcal{C}_{\tilde{y}}|$ denotes the number of words in the document that are indicative of the class. One may find that when $n_c \gg n_s$, $r_{\text{SD}}$ can be quite close to 1 as long as $p$ is large.

**Estimate the best deletion ratio.** We estimate the best deletion ratio for random deletion based on some reasonable assumptions. First, it is easy to see that based on Eq. (2), the optimal deletion ratio is

$$p^* = \left( \frac{n_s}{n_s + n_c} \right)^{\frac{1}{n_c}}, \qquad (3)$$

which depends on both the number of seed words $n_s$ and the number of class-indicative words $n_c$ in the document. For $n_s$, we can simply set it as 1 since the pseudo-label of a document is usually determined by one or two seed words. For $n_c$, we assume that all words in a document are indicative of the true class, except stop words and punctuation. These estimations are acceptable as $p^*$ is almost always close to 1 and is quite robust to the change of $n_s$ and $n_c$ as long as $n_c$ is large (See Figure 2). This condition is likely to be true for realistic datasets (See Table 1). Note that for simplicity, we set one single deletion ratio for a specific dataset. Thus we set $n_c$ as the median number of class-indicative words over all documents in a dataset.

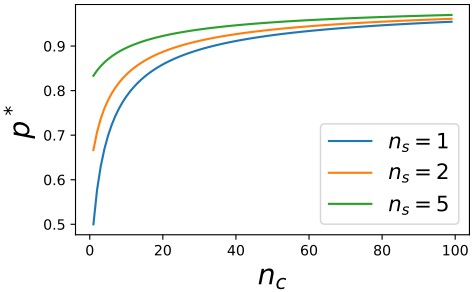

Figure 2: The optimal deletion ratio $p^*$ for random deletion with respect to the number of seed words $n_s$ and the number of class-indicative words $n_c$ based on Eq. (3).

## 4 Experiments

### 4.1 Experiment setup

We evaluate the performance of seed word matching equipped with seed deletion or random deletion on text classification.

**Datasets.** We report the text classification performance on the following datasets, including New York Times (*NYT*), 20 Newsgroups (*20News*), *AG-News* (Zhang et al., 2015), Rotten tomatoes (Pang and Lee, 2005), as well as NYT and 20News datasets with their fine-grained labels respectively. We select these datasets as they cover diverse data properties in text classification, such as topic classification and sentiment analysis, long and short input documents, coarse-grained and fine-grained labels, and balanced and imbalanced label distributions.

For seed word matching, we consider the seed words used in (Mekala and Shang, 2020; Wang et al., 2021). Table 1 shows the statistics of these datasets and the corresponding pseudo-labels given by seed matching.

**Training setting.** We adhere to the following experiment settings for all methods unless otherwise specified. We utilize *BERT* (`bert-base-uncased`) (Devlin et al., 2019) as the text classifier since we found that BERT can produce reliable confidence estimates with our methods in our experiments.

For pseudo-label selection, we select $50\%$ of the high-quality pseudo-labels for all methods and datasets. For random deletion specifically, we set the deletion ratio following the estimation in Section 3.2. The estimated best deletion ratio of each dataset can be found in Figure 3.

Finally, we employ the standard self-training protocol to utilize both the documents labeled by weak supervision and additional documents that are not labeled. Note that if we selected a subset of pseudo-labels before self-training, those pseudo-labeled documents that were not selected will be merged into unlabeled documents. For the self-training process specifically, we train a text classifier on the labeled documents and generate predictions on unlabeled documents. The top $\tau$ fraction of predictions with the highest confidence are then treated as new pseudo-labels and will be merged with the existing pseudo-labels for another training. In our experiments, we conduct this self-training process for 5 iterations.

Note that one can use BERT to select high-quality pseudo-labels alone following our methods while employing advanced text classifiers for subsequent self-training, which may further improve the performance.

**Comparative methods.** We compare our proposed

Table 1: Statistics of the dataset and the corresponding pseudo-labels given by seed matching. For sequence length, we report the median across all pseudo-labeled documents to reduce the impact of outliers, along with the median absolute deviation in the bracket.

| Dataset | # Docs | # Labels | # Pseudo-labeled Docs | Pseudo-label Noise Rate (%) | Sequence Length (Pseudo-labeled Docs) | $n_c$ (Estimated) (Pseudo-labeled Docs) |
|---|---|---|---|---|---|---|
| AGNews | 120,000 | 4 | 32,359 | 16.26 | 39(6) | 27(4) |
| 20News-Coarse | 17,871 | 5 | 7,671 | 12.50 | 286(136) | 140(65) |
| NYT-Coarse | 13,081 | 5 | 9,460 | 11.47 | 922(222) | 462(105) |
| 20News-Fine | 17,871 | 17 | 10,455 | 25.67 | 257(120) | 129(58) |
| NYT-Fine | 13,081 | 26 | 8,229 | 31.80 | 940(214) | 467(100) |
| Rotten-Tomatoes | 10,662 | 2 | 990 | 28.38 | 26(8) | 13(4) |

Table 2: Classification performance achieved by vanilla seed matching and seed matching equipped with various pseudo-label selection methods. $^\dagger$ indicates methods that are not fair comparison and are listed only for reference. We conduct each experiment for $5$ times, report the average, and denote the standard deviation in the bracket. We report both Macro-F1 and Micro-F1 for classification accuracy.

| Method | AGNews | | 20News-Coarse | | NYT-Coarse | | 20News-Fine | | NYT-Fine | | Rotten-Tomatoes | |
|---|---|---|---|---|---|---|---|---|---|---|---|---|
| | Mi-F1 | Ma-F1 | Mi-F1 | Ma-F1 | Mi-F1 | Ma-F1 | Mi-F1 | Ma-F1 | Mi-F1 | Ma-F1 | Mi-F1 | Ma-F1 |
| Oracle$^\dagger$ | 86.3(0.3) | 86.2(0.3) | 90.6(0.4) | 90.6(0.4) | 97.0(0.2) | 93.9(0.4) | 81.1(0.2) | 81.1(0.3) | 96.5(0.1) | 92.0(0.4) | 79.6(1.0) | 79.6(1.0) |
| Vanilla | 83.9(0.6) | 83.9(0.6) | 80.5(0.6) | 80.1(0.6) | 87.8(0.2) | 78.4(0.4) | 68.2(0.6) | 69.0(0.7) | 73.0(0.7) | 68.7(0.8) | 71.4(1.2) | 71.3(1.2) |
| Standard Confidence | 82.2(2.1) | 82.0(2.1) | 78.9(1.8) | 80.0(1.5) | 88.3(4.1) | 79.1(2.8) | 64.4(2.2) | 66.6(1.8) | 45.8(0.5) | 58.6(0.3) | 72.2(2.4) | 72.0(2.4) |
| O2U-Net | 79.8(0.5) | 79.8(0.5) | 80.9(0.3) | 78.5(0.2) | 92.9(0.4) | 85.9(0.7) | 71.1(0.4) | 71.2(0.8) | 14.7(10.2) | 8.70(7.3) | **74.1(1.4)** | **74.0(1.4)** |
| LOPS | 79.5(0.9) | 79.5(0.6) | 81.7(1.0) | 80.7(0.4) | **94.6(0.4)** | **88.4(0.5)** | 73.8(0.6) | 72.7(1.0) | 84.3(0.5) | **81.6(0.3)** | 70.4(0.4) | 70.4(0.4) |
| Seed-Deletion | 84.3(0.7) | 84.2(0.7) | **86.4(0.9)** | **86.1(0.8)** | 92.4(1.3) | 85.0(2.0) | 73.7(0.7) | 75.0(0.5) | 81.7(1.5) | 79.4(1.1) | 70.4(1.3) | 70.3(1.3) |
| Random-Deletion | **86.2(0.5)** | **86.1(0.5)** | 84.4(0.9) | 84.8(0.8) | 91.7(1.3) | 83.3(1.8) | **76.3(0.8)** | **76.8(0.7)** | **84.6(1.4)** | 79.6(1.1) | 73.6(4.3) | 73.4(4.5) |
| Paraphrase$^\dagger$ | 85.4(0.3) | 85.4(0.3) | 86.9(1.2) | 86.7(1.2) | 94.0(0.8) | 88.5(0.7) | 75.6(0.6) | 76.8(0.5) | 76.1(4.0) | 74.8(2.4) | 75.4(0.1) | 75.3(0.1) |
| MLM-Replace$^\dagger$ | 85.8(0.1) | 85.8(0.1) | 87.4(0.1) | 87.5(0.2) | 94.5(0.1) | 88.9(0.2) | 73.6(1.0) | 74.8(0.8) | 84.1(0.6) | 80.0(0.4) | 76.7(1.5) | 76.7(1.5) |

method with the following baselines.

- *Vanilla*: Self-training on all the pseudo-labeled provided by seed matching, without pseudo-label selection.
- *Standard confidence*: Train a classifier on all the pseudo-labeled documents and use its confidence score to select a subset of high-quality pseudo-labels.
- *O2U-Net* (Huang et al., 2019): Train a classifier on all the pseudo-labeled documents and use the normalized loss of each document throughout the training as a metric to select pseudo-labels.
- *LOPS* (Mekala et al., 2022): Train a classifier on all the pseudo-labeled documents and use the learning order cached during training to select pseudo-labels. We follow the setting recommended in their paper and set $\tau = 50\%$.
- *Oracle*$^*$: Based on the true labels, we select only those correct pseudo-labels for self-training. Note that this is not a realistic method and is used only for comparison.

Whenever it is necessary to train a classifier to obtain the confidence, we train for $4$ epochs, in line with the setting in LOPS.

## 4.2 Main results

**Seed-based weak supervision.** We present the classification performance achieved by different methods in Table 2. One may find that seed deletion and random deletion can significantly boost the performance of seed matching-based weakly-supervised classification. On some datasets (e.g., AGNews), random deletion can approach the oracle selection with almost no gap. This demonstrates the performance of the simple seed matching-based weak supervision is greatly underestimated.

Seed deletion and random deletion are also on par with or significantly better than other pseudo-label selection methods including those using sophisticated confidence measures such as the normalized loss in O2U-Net and the learning order in LOPS. In fact, seed deletion and random deletion are still using the standard confidence score of a classifier to select pseudo-labels, albeit they first corrupt the pseudo-labeled documents for training the classifier. Nevertheless, the performance improvement compared to the standard confidence score is huge. For example, the improvement on NYT-Fine is as large as ~ $20\%$ in terms of Macro-F1 and ~ $40\%$ in terms of Micro-F1. This demonstrates that confidence-based pseudo-label selection is greatly underestimated.

Table 3: Classification performance of text classification using a variety of weak supervisions. Results on sophisticated weak supervisions are cited from the corresponding paper. [†] indicates that result is reported as accuracy instead of F1-score in the original paper, while we still include it here since the dataset is class-balanced. We neglect Rotten-Tomatoes here since it is not reported in most of the listed papers.

| Method | AGNews | | 20News-Coarse | | NYT-Coarse | | 20News-Fine | | NYT-Fine | |
|---|---|---|---|---|---|---|---|---|---|---|
| | Mi-F1 | Ma-F1 | Mi-F1 | Ma-F1 | Mi-F1 | Ma-F1 | Mi-F1 | Ma-F1 | Mi-F1 | Ma-F1 |
| ConWea | 73.4 | 73.4 | 74.3 | 74.6 | 93.1 | 87.2 | 68.7 | 68.7 | 87.4 | 77.4 |
| X-Class | 82.4 | 82.3 | 58.2 | 61.1 | **96.3** | **93.3** | 70.4 | 70.4 | 86.6 | 74.7 |
| LOTClass | 84.9 | 84.7 | 47.0 | 35.0 | 70.1 | 30.3 | 12.3 | 10.6 | 5.3 | 4.1 |
| ClassKG | **88.8**[†] | - | 80 | 75 | 96 | 83 | **78** | **77** | **92** | **80** |
| LIME | 87.2 | **87.2** | 79.7 | 79.6 | - | - | - | - | - | - |
| Seed Deletion | 84.3 | 84.2 | **86.4** | **86.1** | 92.4 | 85.0 | 73.7 | 75.0 | 81.7 | 79.4 |
| Random Deletion | 86.2 | 86.1 | 84.4 | 84.8 | 91.7 | 83.3 | 76.3 | 76.8 | 84.6 | **79.6** |

**Compare with sophisticated weak supervision.** In Table 3, we compare the performance of seed word matching equipped with seed word deletion or random deletion with those methods using sophisticated weak supervision sources, listed as follows.

- *ConWea* (Mekala and Shang, 2020) uses pretrained language models to contextualize the weak supervision in an iterative manner.
- *X-Class* (Wang et al., 2021) learns class-oriented document representations based on the label surface names. These document representations are aligned to the classes to obtain pseudo labels.
- *LOTClass* (Meng et al., 2020) obtains synonyms for the class names using pretrained language models and constructs a category vocabulary for each class, which is then used to pseudo-label the documents via string matching.
- *ClassKG* (Zhang et al., 2021) constructs a keyword graph to discover the correlation between keywords. Pseudo-labeling a document would be translated into annotating the subgraph that represents the document.
- *LIME* (Park and Lee, 2022) combines seed matching with an entailment model to better pseudo-label documents and refine the final model via self-training.

For each method, we report the results obtained directly from the corresponding paper. If the results on some datasets are not reported in the original paper, we cite the results in follow-up papers if there are any. We found that with seed deletion or random deletion, seed word matching can be almost as good as or even better than those sophisticated weak supervisions. We thus believe seed word matching can be a competitive baseline and should be considered when developing more complicated weak supervisions.

**Alternative seed-agnostic debiasing methods.** We explore alternative methods to delete the seed words and mitigate the label bias in seed matching-based weak supervision, without knowing the seed words. We consider the following alternatives.

- *MLM-replace*: We randomly mask a subset of words in the document and use BERT to predict the masked words. The document is then corrupted by replacing the masked words with the predictions. This follows the idea of random deletion to delete the seed words probabilistically. Such a method is widely used in other applications (Gao et al., 2021; Clark et al., 2020).
- *Paraphrase*: We generate the paraphrase of a document using the T5 model (Raffel et al., 2019) fine-tuned on a paraphrase dataset APWS (Zhang et al., 2019). We use the publicly available implementation at (Duerr, 2021). This is a straightforward method to delete the seed words.

Since these alternative methods only serve as a reference for our main methods, we search their best hyperparameter, namely the mask ratio for MLM-replace and the token-generation temperature for paraphrase respectively. As shown in Table 2, these alternative methods can work as well as or better than random deletion. However, in practice, we would prefer using random deletion since it requires no extra model or knowledge source.

### 4.3 Study on random deletion

**Deletion ratio in random deletion.** We verify whether our estimation of the best deletion ratio is reasonable. In Figure 3, we modulate the deletion ratio and check the classification performance of random deletion for different datasets. One can find that as the deletion ratio increases, the performance first increases and then decreases, which is aligned

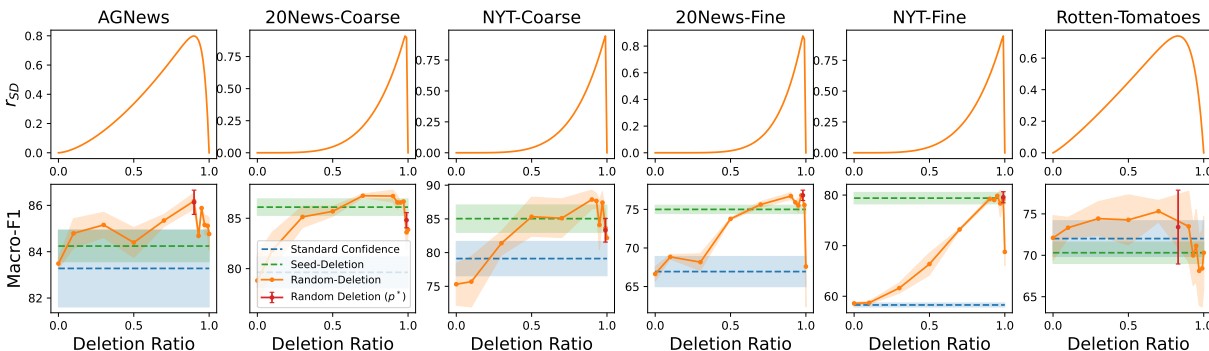

Figure 3: (*Top*) The seed-deletion rate $r_{\text{SD}}$ given different deletion ratios (Eq. (2)), where $n_s$ and $n_c$ are estimated for each dataset as mentioned in Section 3.2. (*Bottom*) The classification performance of random deletion given different deletion ratios. "($p^*$)" denotes the one using the best deletion ratio estimated for each dataset. We also denote the performance of seed deletion and standard confidence for comparison.

with the trend of the seed-deletion rate $r_{\text{SD}}$ analyzed in Section 3.2. The performance peaks when the deletion ratio is large ($\gtrsim 0.9$), which matches our estimation of the best deletion ratio. Furthermore, one may find that the best deletion ratio is relatively smaller for datasets with a shorter sequence length (e.g., AGNews and Rotten-Tomatoes), compared to that for datasets with a long sequence length (e.g., 20News and NYT), which is also predicted by our estimation.

**How does random deletion work?.** One may notice that in Table 2, random deletion can outperform seed deletion on a few datasets. This indicates that random deletion has an additional regularization effect on top of deleting seeds, potentially due to more intense data augmentation.

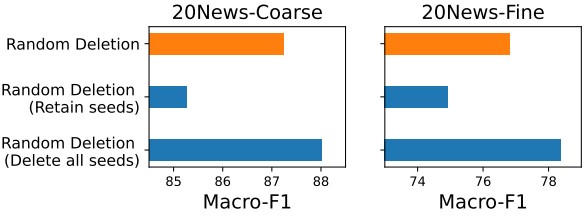

Figure 4: Classification performance achieved by variations of random deletion with seed words always retained in the document ("Retain seeds") and with seed words always deleted completely ("Delete all seeds").

However, we note that random deletion works not entirely because of this additional regularization effect. To show this, we conduct ablation experiments with two additional methods. The first is random deletion but with the seed words always retained in the document. The second is random deletion but with seed words first deleted completely. For a fair comparison, we search the best deletion ratio for these different methods including the

standard random deletion. We experiment on two representative datasets including 20News-Coarse and 20News-Fine due to computational constraints.

Figure 4 shows that when seed words are always retained, random deletion achieves significantly worse performance than the standard random deletion, although the former is merely deleting one or two words fewer. On the other hand, when seed words are already deleted, further random deletion only slightly improves the performance compared to the standard random deletion. These pieces of evidence demonstrate that the benefit of random deletion may be partly attributed to a regularization effect, but the deletion of seeds and thus the mitigation of label bias is still one important factor.

**Compare with additional regularization methods.** Since random deletion may introduce an ad-

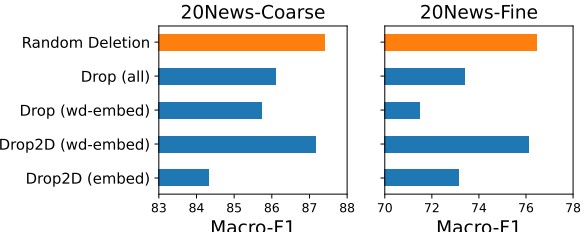

Figure 5: Classification performance using dropout as a regularization method for pseudo-label selection. We try two types of dropout including Dropout ("Drop") and Dropout-2D ("Drop2D"). We try dropout at various positions in the transformer architecture, including all dropouts layers in the original transformer ("all"), the embedding layer only ("embed"), and the word embedding layer only ("wd embed").

ditional regularization effect, we compare it with other regularization methods that can potentially reduce the label bias. We will mainly compare different types of dropout (see below) that are widely

Table 4: Original document versus corrupted document after random deletion, along with the corresponding pseudo-label and true label. Here the example documents are randomly picked from the pseudo-labeled data in the 20News-Coarse dataset.

| | | **Original Document** | **Corrupted Document**
(Random Deletion) |
|---|---|---|---|
| **Pseudo-label** | Computer | re increasing the number of serial ports distribution world nntp posting host mac4. jpl. nasa. gov in article ( steven langlois ) wrote does anyone know if there are any devices available for the **mac** which will increase the number of serial ports available for use simultaneously? i would like to connect up to 8 serial devices to my **mac** for an application i am working on. ... | distribution posting4 the available application working must independently. such any to the serial? ink bug only system the. them then of the. to using |
| **Seed Word** | mac | | |
| **True-label** | Computer | | |
| **Pseudo-label** | Computer | re dumb options list in article ( charles parr ) writes the idea here is to list pointless options. you know, stuff you get on a car that has no earthly use? 1 ) power **windows** i like my power **windows**. i think they're worth it. however, cruise control is a pretty dumb option. what's the point? if you're on a long trip, you floor the gas and keep your eyes on the rear view mirror for cops, right? power seats are pretty dumb too, unless you're unlucky enough to have to share your car. ... | the options you that ).? keep are enough have like " do paper. breath. ' |
| **Seed Word** | windows | | |
| **True-label** | Sports | | |
| **Pseudo-label** | Science | re pro abortion feminist leader endorses trashing of free speech rights in article ( gordon fitch ) writes ( doug holtsinger ) writes 51 arrested for defying judge's order at abortion protest rally the miami herald, april 11, 1993 **circuit** judge robert mcgregor's order prohibits anti abortion pickets within 36 feet of the property line of aware woman center for choice. even across the street, they may not display pictures of dead fetuses or sing or chant loud enough to be heard by patients inside the clinic. ... | leader trashying judge 11 judge abortion the display pictures loud as similar an appeal group from have a rock and then see the he homes did speech of the hear there from to expression on particular information, to others if be considered the'ists arson to else the |
| **Seed Word** | circuit | | |
| **True-label** | Politics | | |

used in classification tasks, since it is mostly similar to random deletion. We defer other commonly seen regularization methods to the appendix. We consider applying dropout to different positions in the transformer model. For a fair comparison, we search the best dropout ratio for these dropout methods, as well as the best deletion ratio for our random deletion. We experiment on two representative datasets due to computational constraints.

- *Dropout* (Hinton et al., 2012) is a classic regularization method to prevent overfitting. We simply utilize the dropout layers built in the original transformer architecture, including those after the embedding layers, self-attention layers, and feed-forward layers, following previous work (Gao et al., 2021).
- *Dropout-2D* (Tompson et al., 2014) is different from vanilla Dropout in that it drops the entire channel as a whole. We only apply this to the embedding layer in the transformer to drop the entire embedding of a word or a position.

As shown in Figure 5, random deletion consistently outperforms other regularization methods. The only regularization method that can compete with random deletion is Dropout-2D applied on the word embedding layer specifically. However, one may note that this dropout variation is in fact almost equivalent to random deletion since the entire embedding of a word will be randomly dropped. These again demonstrate that random deletion works not simply because of a regulariza-

tion effect.

**Case study.** We manually inspect the pseudo-labeled documents after random deletion to see if the label bias can be mitigated. In Table 4, we randomly pick some example documents after random deletion and find that the seed words are indeed deleted and some class-indicative words are still present to allow effective classification.

## 5 Conclusion and Future Work

In this paper, we revisit the simple seed matching-based weakly supervised text classification method and show that if its pseudo-labels are properly debiased, it can achieve state-of-the-art accuracy on many popular datasets, outperforming more sophisticated types of weak supervision. Specifically, our controlled experiments show that confidence-base selection of seed matching-based pseudo-labels is ineffective largely because of the label bias injected by the simple, yet erroneous seed-match rule. We propose two effective debiasing methods, seed deletion, and random deletion, which can mitigate the label bias and significantly improve seed matching-based weakly supervised text classification.

In future work, we plan to extend this debiasing methodology to broader problems and methods. For example, for weakly supervised text classification, we wish to explore a generalization of the debiasing strategy to more sophisticated types of weak supervision. It will also be interesting to develop a backdoor defense framework around the

proposed methods, especially random deletion.

## Limitations

We have shown that randomly deleting the words work well for seed-matching-based weak supervision without knowing the seed words. However, this idea might not generalize straightforwardly to more sophisticated types of weak supervision. We have tried to apply random deletion to X-Class, but have not observed a significant improvement in the text classification performance. We hypothesize that this is because the pseudo-labels in X-Class are not generated based on seed word matching, but rather based on the similarity between the label embeddings provided by pretrained language models. We believe a label debiasing method universally applicable to all types of weak supervision is still under-explored.

## Ethical Consideration

This paper analyzes the difficulty of identifying label errors in the pseudo-labels generated by simple seed-matching weak supervision. This paper proposed methods to better identify such label errors and improve weakly supervised text classifiers. We do not anticipate any major ethical concerns.

## Acknowledgements

We thank the anonymous reviewers for their helpful feedback. Our work is sponsored in part by NSF CAREER Award 2239440, NSF Proto-OKN Award 2333790, NIH Bridge2AI Center Program under award 1U54HG012510-01, Cisco-UCSD Sponsored Research Project, as well as generous gifts from Google, Adobe, and Teradata. Any opinions, findings, and conclusions or recommendations expressed herein are those of the authors and should not be interpreted as necessarily representing the views, either expressed or implied, of the U.S. Government. The U.S. Government is authorized to reproduce and distribute reprints for government purposes not withstanding any copyright annotation hereon.

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

## A Appendix

**Effect of selection ratio.** As mentioned before, our methods do not introduce additional hyperparameters. Nevertheless, for pseudo-label selection in general, the selection fraction could be an important hyperparameter as the noise rate in the selected subset of pseudo-labels can vary significantly if we select different fractions, as also shown in Figure 1. Therefore, we check the performance of the standard confidence-based pseudo-label selection and the confidence-based selection equipped with seed deletion and random deletion, as the selection fraction varies. Figure 6 shows that our proposed methods are consistently better than standard confidence and achieve relatively robust performance as the selection fraction varies. The performance peaks when the selection fraction is moderate (~ 50%) for different datasets.

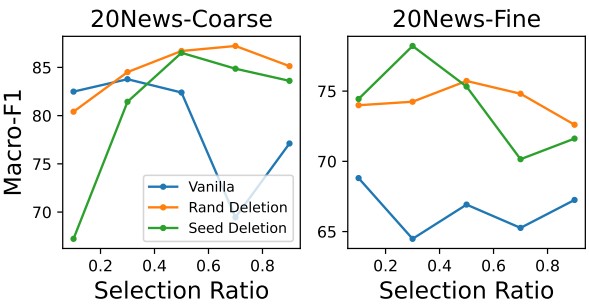

Figure 6: The classification performance when selecting the pseudo-labels at different fractions.

**Experiments on additional confidence regularization methods.** We experiment on additional methods for regularizing the confidence learning of the text classifier, including the following.

- *MC-Dropout* (Gal and Ghahramani, 2015) randomizes the network inference process by dropping intermediate activations. The average output over multiple inferences is utilized as a more reliable confidence score.

- *Early stopping* is often utilized as a regularization to help mitigate overfitting. Empirically, it is observed that an early-stopped model is less prone to learning noisy data (Arpit et al., 2017).

Here, we treat each method as a baseline and compare it with the corresponding method combined with random deletion. For each method, we modulate its most important hyperparameter, namely number of passes for MC-Dropout and number of

training epochs for early stopping respectively. As shown in Figures 7 and 8, random deletion consistently outperforms the baseline for different confidence regularization methods. Random deletion also achieves more robust performance across different hyperparameter settings, which is important for weakly-supervised classification since we often lack a large clean dataset to select the best hyperparameter.

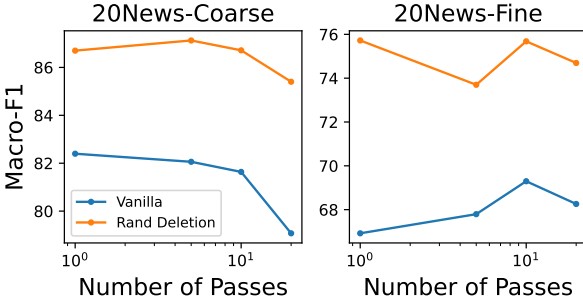

Figure 7: Classification performance using MC-dropout to obtain better confidence for pseudo-label selection. We test different numbers of passes for MC-dropout.

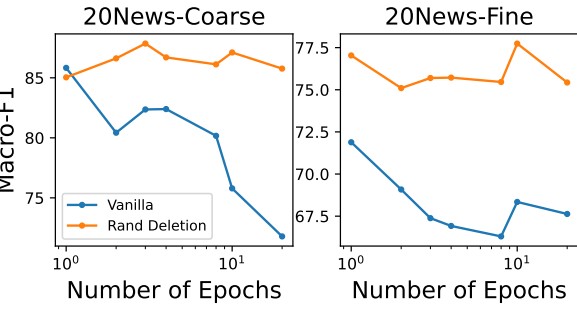

Figure 8: Classification performance using early stopping as a regularization method to obtain better confidence for pseudo-label selection. We check the performance when early stopping at different epochs.