# OpenReview forum: "Debiasing Made State-of-the-art: Revisiting the Simple Seed-based Weak Supervision for Text Classification"
_EMNLP/2023/Conference — EMNLP 2023 Main_

### Official Review · Reviewer_Lv81 · 2023-07-21

**Soundness:** 4

**Excitement:**

4: Strong: This paper deepens the understanding of some phenomenon or lowers the barriers to an existing research direction.

**Paper Topic And Main Contributions:**

The paper addresses the task of weakly supervised text classification, specifically when employing seed word lists per class to generate pseudo-labels. Naively training a SSL model on these pseudo-label may result in the classifier learning a “biased”, simple correlation between those seed words and the class label, which does not generalize well. The authors show that this seed bias can be overcome by deleting seed words from the document before training. In addition, they show that a random deletion strategy works even better when a rather larger fraction of words per document is deleted. Experiments on 6 text classification datasets show that both deletion approaches can achieve SotA accuracy, compared to more sophisticated pseudo-labeling strategies.

**Questions For The Authors:**

- Question A: What are the actual deletion ratios used for the 6 datasets in Table 3 (maybe I overlooked this info?)?
- Question B: Since the deletion ratio p seems to be quite high at p>= 0.9 according to Sec 4.3, this means that per document <= 10% of the class-indicative tokens are retained, plus punctuation. If a given class contains >10 documents, I’d assume due to the random/uniform sampling, each class-indicative word may still occur in one of the docs. What happens in the case of very long-tail classes with e.g. <10 documents - in this case, at least some of the class-indicative words are lost since they get deleted in all documents, correct? How is the performance in this case? And also - for very long documents, doesn’t deleting so many class-indicative words mean that the ratio of punctuation (non-informative) to words (informative) is very skewed towards the former? Does that affect classification performance? (Or maybe I've misunderstood the deletion procedure?)

**Reasons To Accept:**

- paper is well organized and easy to follow
- very interesting analysis on "debiasing" seed-based pseudo-labeling
- experimental results are convincing

**Reasons To Reject:**

- I can’t find any

**Reproducibility:**

4: Could mostly reproduce the results, but there may be some variation because of sample variance or minor variations in their interpretation of the protocol or method.

**Reviewer Confidence:**

2: Willing to defend my evaluation, but it is fairly likely that I missed some details, didn't understand some central points, or can't be sure about the novelty of the work.

---

> ### Author Rebuttal · Authors · 2023-08-29
>
> Thank you for the valuable comments. Please see our reponse below.
>
> ## ____
> __Q. Clarification of the deletion ratio__
>
> Throughout the paper, we determine the deletion ratio of random deletion based on our estimation in Section 3.2 (Equation (3)). One can find the estimated deletion ratio for each dataset in Figure 3. We are sorry for the confusion and will make it clear in the revision.
>
> ## ____
> __Q. Performance on long-tail classes and long documents__
>
> First, we note that our random deletion is not static but performed along with training. This means that for classes with very few documents (long-tail classes), the classifier can still see all the class-indicative words as long as the training is conducted for sufficiently many epochs.
> One can also employ standard strategies for dealing with highly imbalanced datasets (e.g., upsampling), where random deletion should properly fit in, since the seed-deletion rate (i.e., Equation (1), the probability that any training document contains no seed word but at least one class-indicative word) remains the same.
>
> Second, for long documents, the ratio between class-indicative words and non-class-indicative words should remain approximately the same when we perform random deletion, even for a very high deletion ratio. This is because the deleted words are uniformly sampled, which means the class-indicative words and non-class-indicative words have an equal chance of being deleted.
>
> Finally, we would like to emphasize that the performance of the text classifier when training with random deletion is not a major concern. This is because random deletion (as well as seed deletion) is designed to improve the confidence estimates of the text classifier, such that they can be used to select the correct pseudo-labels. After the high-quality pseudo-labels are selected, one can use any classifier to achieve the best performance.
>
> In fact, existing works have shown that the performance of classification models is usually orthogonal to its confidence estimates [1]. In our experiments, we also observe that the classifier used to select the pseudo-labels often has low accuracy but well-calibrated confidence estimates.
>
> ## ____
> __Reference__
>
> [1] On Calibration of Modern Neural Networks. Guo et al., 2017.

---

### Official Review · Reviewer_wbLQ · 2023-07-27

**Typos Grammar Style And Presentation Improvements:** N/A
**Soundness:** 4

**Excitement:**

4: Strong: This paper deepens the understanding of some phenomenon or lowers the barriers to an existing research direction.

**Missing References:**

N/A

**Paper Topic And Main Contributions:**

The aim of the paper is to improve the performance of weakly supervised text classifiers by revisiting the seed matching-based method for generating pseudo-labels. Prelimiary study presented in the paper shows that the limited performance of seed matching-based methods is due to a bias caused by the seed words which prevents the classifier from learning reliable confidence for high quality pseudo labels. The authors propose two simple techniques for overcoming this problem, based on seed word deletion and random deletion. The seed word deletion method is based on deleting the seed words from the the pseudo-labeled texts. The random deletion method aims to handle cases where the seed words are unknown and it is based on random token deletion in a probablistic manner. Both methods are shown to mitigate the label bias problem caused by the seed words and help improve the performance of the classifier. For their experiments, the authors used 4 datasets (New York Times, 20 Newsgroups, AG-News, and Rotten tomatoes) and utilise BERT as text classifier. The authors compared the proposed methods with a wide range of seed-based weak supervision approaches as well as more sophisticated weak suprivision techniques. Results showed that the proposed methods outperform other seed-based approaches and also perform on par  with or even better than more sophisticated techniques.

**Questions For The Authors:**

1) Why did you select BERT? Do you expect significant changes in the results if a different classifier is used?

**Reasons To Accept:**

The novelty and the contributions of the paper are very well justified. The analyses are extensive and show a clear advantage of the method over exsiting approaches, including state-of-the-art. The methods are well described and easy to implement/adjust to wider range of the datasets and classification tasks.

**Reasons To Reject:**

The use of a sinlge classifier - BERT. The results might differ if different classifiers are used.

**Reproducibility:**

3: Could reproduce the results with some difficulty. The settings of parameters are underspecified or subjectively determined; the training/evaluation data are not widely available.

**Reviewer Confidence:**

4: Quite sure. I tried to check the important points carefully. It's unlikely, though conceivable, that I missed something that should affect my ratings.

---

> ### Author Rebuttal · Authors · 2023-08-29
>
> Thank you for the valuable comments. Please see our reponse below.
>
> ## ____
> __Q. BERT as the text classifier__
>
> We believe BERT is arguably the most widely used base model for text classification. Existing works on weakly-supervised text classification also often employ BERT (or its variation) as the single base classifier to demonstrate the results, e.g., LOTClass [1] and X-Class [2].
>
> Furthermore, we expect our methods can still help for other base classifiers.
> This is because our methods are specifically designed to better select the correct weak pseudo-labels, by leveraging the confidence estimates of a model. In our experiments, we found that BERT can produce reliable confidence estimates with our methods.
> Once the subset of pseudo-labels is determined, one can in fact train with any text classifier, which should achieve consistently better classification performance with higher-quality pseudo-labels.
>
> ## ____
> **Reference**
>
> [1] Text Classification Using Label Names Only: A Language Model Self-training Approach. Meng et al., 2020.
>
> [2] X-class: Text Classification With Extremely Weak Supervision. Wang et al., 2021.

---

### Official Review · Reviewer_PSwo · 2023-08-05

**Soundness:** 4

**Excitement:**

4: Strong: This paper deepens the understanding of some phenomenon or lowers the barriers to an existing research direction.

**Paper Topic And Main Contributions:**

In this paper the authors propose an improvement for the seed matching-based method for weak supervision. They define the label bias injected by the seed-matching rule and argue that this is the cause of the poor performance of the method. They implement two methods to delete seed words from pseudo-labeled texts and show that they improve the results.

**Reasons To Accept:**

- The proposed method although simple, is very interesting and shows competitive results or even outperforms other more complicated methods.
- The authors provide an extensive study about how the deletion of seed words should be done and perform experiments that give insights on how the random deletion works and the decision of the deletion ratio.


**Reasons To Reject:**

The proposed method has lower scores than its competitors in 3 out of the five datasets used for evaluation.

**Reproducibility:**

3: Could reproduce the results with some difficulty. The settings of parameters are underspecified or subjectively determined; the training/evaluation data are not widely available.

**Reviewer Confidence:**

2: Willing to defend my evaluation, but it is fairly likely that I missed some details, didn't understand some central points, or can't be sure about the novelty of the work.

---

> ### Author Rebuttal · Authors · 2023-08-29
>
> Thank you for the valuable comments. Please see our reponse below.
>
> ## ____
> __Q. Clarification of the results in Table 3__
>
> We assume the reviewer is referring to the results reported in Table 3. However, note that here the comparison between the competitive methods and ours is not entirely fair, since they often employ different base classifiers and/or additional datasets. For example, ClassKG [1] uses Longformer [2] for the classification of long-text datasets, which include the 20News-groups and New York-Times datasets (4 out of 5 datasets we used in our experiments). And LIME [3] employs BART-large [4] as the model to select pseudo-labels. They also fine-tune the model on MultiNLI [5] before being used for pseudo-label selection.
>
> Nevertheless, our method can still achieve results that are quite close to these methods. On 4 out of 5 datasets, the gap between our result and the best existing result is less than or equal to 1 (Macro-F1). We believe this demonstrates that our method can be on par with the state-of-the-art while being significantly simpler.
>
> ## ____
> __Reference__
>
> [1] Weakly-supervised Text Classification based on Keyword Graph. Zhang et al., 2021.
>
> [2] Longformer: The Long-document Transformer. Beltagy et al., 2020.
>
> [3] Lime: Weakly-supervised text classification without seeds. Park et al., 2022.
>
> [4] BART: Denoising Sequence-to-sequence Pre-training for Natural Language Generation, Translation, and Comprehension. Lewis et al., 2020.
>
> [5] A Broad-coverage Challenge Corpus for Sentence Understanding through Inference. Williams et al., 2018.

---

### Meta-Review · Area_Chair_cWix · 2023-09-14

**Recommendation:** 5

**Metareview:**

This work focuses on the problem of weakly labeled data for text classification and proposes a simple yet effective method. Reviewers unanimously applauded the motivation, presentation, and novelty of the work. Only minor issues were regarding breadth of experiments and missing references, none of which detract from the quality of the work.

---

### Decision · Program_Chairs · 2023-10-07

**Decision:**

Accept-Main

**Comment:**

This work focuses on the problem of weakly labeled data for text classification and proposes a simple yet effective method. Reviewers unanimously applauded the motivation, presentation, and novelty of the work. Only minor issues were regarding breadth of experiments and missing references, none of which detract from the quality of the work.